# Bronchoalveolar Lavage Fluid from COPD Patients Reveals More Compounds Associated with Disease than Matched Plasma

**DOI:** 10.3390/metabo9080157

**Published:** 2019-07-25

**Authors:** Eitan Halper-Stromberg, Lucas Gillenwater, Charmion Cruickshank-Quinn, Wanda Kay O’Neal, Nichole Reisdorph, Irina Petrache, Yonghua Zhuang, Wassim W. Labaki, Jeffrey L. Curtis, James Wells, Stephen Rennard, Katherine A. Pratte, Prescott Woodruff, Kathleen A. Stringer, Katerina Kechris, Russell P. Bowler

**Affiliations:** 1School of Medicine, University of Colorado, Aurora, CO 80045, USA; 2Pathology Department, Johns Hopkins University, Baltimore, MD 21287, USA; 3Department of Medicine, National Jewish Health, Denver, CO 80206, USA; 4Agilent Technologies, Santa Clara, CA 95051, USA; 5Department of Marsico, Lung Institute/Cystic Fibrosis Center, University of North Carolina at Chapel Hill, Chapel Hill, NC 27599, USA; 6Skaggs School of Pharmacy and Pharmaceutical Sciences, University of Colorado Anschutz Medical Campus, Aurora, CO 80045, USA; 7Department of Biostatistics, Colorado School of Public Health, Aurora, CO 80045, USA; 8Division of Pulmonary and Critical Care Medicine, University of Michigan, Ann Arbor, MI 48109, USA; 9Division of Pulmonary Biology, Cincinnati Children’s Hospital Medical Center, Cincinnati, OH 45229, USA; 10BioPharmaceuticals R&D, AstraZeneca, Cambridge CB4 0XR, UK; 11Department of Internal Medicine, University of Nebraska Medical Center, Omaha, NE 68588, USA; 12Department of Medicine, UCSF Pulmonary, Critical Care, Allergy and Sleep Medicine, University of California, San Francisco, CA 94143, USA; 13Department of Clinical Pharmacy, College of Pharmacy, University of Michigan, Ann Arbor, MI 48109, USA

**Keywords:** metabolomics, COPD, emphysema, mass spectrometry, LC–MS, bronchoalveolar lavage, BAL, BALF, plasma

## Abstract

Smoking causes chronic obstructive pulmonary disease (COPD). Though recent studies identified a COPD metabolomic signature in blood, no large studies examine the metabolome in bronchoalveolar lavage (BAL) fluid, a more direct representation of lung cell metabolism. We performed untargeted liquid chromatography–mass spectrometry (LC–MS) on BAL and matched plasma from 115 subjects from the SPIROMICS cohort. Regression was performed with COPD phenotypes as the outcome and metabolites as the predictor, adjusted for clinical covariates and false discovery rate. Weighted gene co-expression network analysis (WGCNA) grouped metabolites into modules which were then associated with phenotypes. K-means clustering grouped similar subjects. We detected 7939 and 10,561 compounds in BAL and paired plasma samples, respectively. FEV_1_/FVC (Forced Expiratory Volume in One Second/Forced Vital Capacity) ratio, emphysema, FEV_1_ % predicted, and COPD exacerbations associated with 1230, 792, eight, and one BAL compounds, respectively. Only two plasma compounds associated with a COPD phenotype (emphysema). Three BAL co-expression modules associated with FEV_1_/FVC and emphysema. K-means BAL metabolomic signature clustering identified two groups, one with more airway obstruction (34% of subjects, median FEV_1_/FVC 0.67), one with less (66% of subjects, median FEV_1_/FVC 0.77; *p* < 2 × 10^−4^). Associations between metabolites and COPD phenotypes are more robustly represented in BAL compared to plasma.

## 1. Introduction

Chronic obstructive pulmonary disease (COPD) prevalence is 6% in the United States and caused roughly 700,000 hospitalizations, 1.5 million emergency room visits, and 10 million physician visits in 2010 [1]. Although airflow obstruction by spirometry is the sine qua non of research definitions of COPD, there are phenotypes of COPD such as emphysema, chronic bronchitis, and frequent exacerbators [2] that spirometry alone does not distinguish [3]. New technologies such as whole genome sequencing, transcriptomics, proteomics, and metabolomics could provide much needed insight into the molecular mechanisms that underlie these phenotypes and fulfill goals consistent with personalized medicine [3]. 

Similar to other complex diseases, COPD “-omics” studies have largely focused on DNA (genetic) and RNA (transcriptomic) signatures [4]. Recent technological developments are making high throughput mass spectrometry (MS) proteomics and metabolomics more feasible for large cohort research [5] and blood metabolomic COPD signature compounds include sphingolipids and amino acids [6,7].

A recent study of serum from 4742 subjects from the Atherosclerosis Risk in Communities (ARIC) cohort replicated the previous observation that amino acid metabolism is associated with COPD [8]. Interestingly, though this study highlighted compounds involved in amino acid metabolism, it also suggested areas of weakness in using blood to study COPD. More compounds were found to be associated with FEV_1_ and FVC than FEV_1_/FVC. The authors suggested this may indicate greater ability to detect lung size as opposed to obstructive airflow associations with serum compounds. Also, the change in concentration of the branched chain amino acids, in particular, with respect to phenotype, was inconsistent with other reports. Most previous studies, like ARIC, have used blood for metabolomics assays even though the first line of exposure to tobacco smoke is the lung epithelial lining fluid. The main limitations to obtaining bronchoalveolar lavage (BAL) are the procedural risks and higher cost compared with blood. In this study, we acquired BAL and contemporaneous plasma samples from 115 subjects, with or without COPD, enrolled in the Subpopulations and Intermediate Outcomes in COPD Study (SPIROMICS) and analyzed them using untargeted liquid chromatography (LC)–mass spectrometry (MS) metabolomics. To our knowledge, this is the largest study of its kind to use this approach for the purpose of detecting metabolites in two biofluids associated with COPD.

## 2. Results

### 2.1. Cohort Characteristics

Characteristics of the cohort are displayed in Table 1. For comparison, the SPIROMICS cohort characteristics are shown in Appendix A.

### 2.2. Compounds Detected in BAL and Plasma

In both BAL and plasma, 15,019 compounds were detected (Figure 1). Most of these compounds were unique to either BAL (4458, 30%) or plasma (7080, 47%), with compounds detected in both amounting to 3481 (23%, Figure 1A). Annotation was only available for 5866 (39%) compounds (Figure 1B). Of the named compounds, 2058 (35%) were detected in both BAL and plasma, representing 65% and 43% of the compounds from each fluid, respectively (Figure 1B). HMDB (Human Metabolome Database) and KEGG (Kyoto Encyclopedia of Genes and Genomes) IDs were available for 9% and 2% of compounds, respectively (Figure 1C,D). Median (IQR) correlation between BAL and plasma compounds was 0.02 (−0.09, 0.13). For annotated compounds, median (IQR) of correlation was 0.02 (−0.05, 0.09) (Figure 2).

### 2.3. Compound Associations with Clinical Covariates

Of the four clinical variables used as covariates, current smoking status had the most significant associations with compounds in BAL. Sex and age had the most significant associations with compounds in plasma (Table 2, Appendix A). 

### 2.4. Compound Associations with Plasma Cell-Counts

Of the five plasma-based cell-count phenotypes, neutrophil count had the most significant associations (665) with compounds in BAL. Hemoglobin and hematocrit had the most significant associations with compounds in plasma (Table 2, Appendix A).

### 2.5. Compound Associations with BAL Cell-Counts

Of the five BAL-based cell-count phenotypes, lymphocyte count, monocyte count, and macrophage count were each associated with one BAL compound. Eosinophil count, neutrophil, and macrophage count were associated with seven, four, and one compounds in plasma, respectively (Table 2, Appendix A). BAL-based cell-count phenotypes yielded much fewer compound associations than plasma-based cell-count phenotypes.

### 2.6. Compound Associations with COPD Phenotypes

For the COPD phenotypes, 1361 compounds in BAL were associated with at least one of the five phenotypes, with FEV_1_/FVC containing most of the total (1230, 90%). Percent emphysema, FEV_1_ % predicted, and exacerbations/yr were associated with 792, eight, and one compounds, respectively. In plasma, two compounds were associated with percent emphysema (Table 2, Appendix A).

### 2.7. Compounds Most Highly Associated With Spirometry

Excluding unannotated compounds for which no interpretation was performed, compounds most strongly associated with FEV_1_/FVC included one nicotine metabolite, p-cresol, four phosphatidylethanolamines, four phosphatidylcholines, two cardiolipins, free homocysteine, one bile acid, one sphingolipid, one cysteine derived compound, one glycine derived compound, one threonine derived compound, one sphingomyelin, two glycerolipids, and two likely xenobiotics (Table 3). 

### 2.8. Significantly Enriched Compound Classes

The set of BAL FEV_1_/FVC-associated compounds was significantly enriched for multiple classes of molecules, including amino acid derived compounds, fatty acids, and phospholipids including phosphatidylethanolamines, lysophosphatidylethanolamines, lysophosphatidylcholines, phosphatidylserines, phosphatidylinositols, and phosphatidylcholines (Figure 3A). The set of BAL emphysema-associated compounds was significantly enriched for the same categories excluding lysophospholipids and phosphatidylserines and with the addition of carnitine containing compounds (Figure 3C). Within the amino acid derived compounds, amino acids most significantly associated with FEV_1_/FVC included arginine, isoleucine, and serine (Figure 3B). Amino acid derived compounds significantly associated with emphysema included leucine and lysine. (Figure 3D). The direction of association for significant amino acids followed the same pattern as the overall amino acid derived compound category, decreased with decreasing FEV_1_/FVC ratio and increasing emphysema.

### 2.9. Co-Expressed BAL Compounds Grouped into Modules Associated with COPD Phenotypes

Weighted gene co-expression network analysis (WGCNA) identified 12 modules of co-abundant compounds in BAL and 30 modules of co-abundant compounds in plasma, not including the “grey” module reserved for compounds that could not be clustered. The largest module identified in BAL, comprising 1339 compounds, was also the module most significantly correlated with COPD phenotypes, including FEV_1_/FVC, % emphysema, chronic bronchitis, and FEV_1_ % predicted (Figure 4A). The compounds populating this module overlapped with the compounds found to associate with FEV_1_/FVC and % emphysema using regression analysis (Appendix A, Jaccard Similarity Index 0.60 and 0.52, respectively). Compounds within this module most tightly correlated with the eigenvector of the module (i.e., hub compounds) were also individually associated with FEV_1_/FVC. One hundred percent of the 300 most correlated compounds with the module eigenvector were also individually associated with FEV_1_/FVC.

In BAL, age and current smoking status were also significantly correlated with modules of sizes 57 and 33, respectively. The largest module identified in plasma, comprising 4279 compounds, was the module of non-co-abundant compounds, the grey module (Appendix A). Correlation between COPD phenotypes and compound modules was lower in plasma than in BAL but higher for sex, age, BMI (Body Mass Index), and hemoglobin, corresponding to modules of sizes 42, 147, 263, and 78, respectively (Figure 4B, Appendix A). Of BAL cell counts, BAL eosinophils correlated most highly with a BAL compound module (Appendix A).

### 2.10. Grouping on Compound Profile Separated People with Differing Lung Function

In BAL, subject level clustering based on K-means of Euclidian distance between profiles of all compounds demonstrated two subject groups, one with relatively decreased lung function and one with relatively preserved lung function, using spirometry as a surrogate for lung function (Appendix A). Silhouette scores were used to choose the optimal cluster number of two. Scores were generated for two to nine clusters. Highest mean silhouette score was for two clusters (0.30) with the next highest silhouette score for three clusters (0.17). In plasma, using the same clustering technique, no association was observed between FEV_1_/FVC and cluster assignment (Appendix A).

## 3. Discussion

To our knowledge, this is the largest reported untargeted LC–MS-based metabolomics analysis performed in COPD BAL fluid. An additional strength is that the same assay was performed on the subjects’ simultaneously obtained plasma. Three important observations followed. First, BAL compounds correlated poorly with plasma compounds, suggesting that BAL and plasma can provide independent biomarker information. Second, BAL had, by far, more compounds associated with COPD phenotypes than plasma, notably FEV_1_/FVC. Third, the BAL compounds associated with FEV_1_/FVC were enriched for multiple compound classes such as: amino acid containing compounds, fatty acids, and phospholipids including lysophospholipids, phosphatidylethanolamines, phosphatidylinositols, phosphatidylcholines, and phosphatidylserines.

There are few published data on BAL small molecule compounds from untargeted mass-spectrometry in COPD. Previous investigations of BAL in lung disease studied small molecules in smokers vs. non-smokers [9], subjects with ARDS (Acute Respiratory Distress Syndrome) [10], or emphysematous mice [11]. Some of the same pathways dysregulated in ARDS, such as fatty acids, amino acids, phospholipids, and phosphatidylcholines, were also significant in our study. We also observed particularly strong associations between current smoking and BAL metabolome (e.g., amino acids and fatty acids). This is similar to a mouse model of emphysema. The mouse model yielded results aligning with results in our study in multiple ways—BAL metabolites yielded more significant differences than plasma metabolites, BAL in emphysema had depleted levels of phosphatidylcholine, lysophosphatidylcholine, amino acids, and carnitine, and BAL metabolites distinguished emphysematous mice from non-emphysematous mice more readily than plasma metabolites.

In our work, the top BAL compounds associated with FEV_1_/FVC ratio included p-cresol, a metabolite of human gut microbiota and nicotine, four phosphatidylethanolamines (a type of phospholipid), free homocysteine, a cysteine containing compound, and a sphingomyelin. Some of these compounds may play a direct causal role in airway obstruction in the lung while others may be only biomarkers.

Possible non-causal compounds include p-cresol and homocysteine. P-cresol has been noted to be toxic in high doses, especially in the context of renal impairment, and is associated with the microbiome. In BAL however, p-cresol may serve as a biomarker for microbiome-lung function interaction rather than directly instigating pathogenesis [12]. Homocysteine, a compound reported previously as elevated in the plasma of COPD subjects (among other diseases), may also be an indirect rather than direct causal player in the disease [13]. Previous studies have not shown that decreasing homocysteine with folic acid dampens inflammatory processes [14].

Compounds involved in oxidative stress, such as cysteine and lysophosphatidylcholine, may serve as causal in lung function decline. Cysteine is involved in anti-oxidant activity [15]. Increased free radical activity and consequent inflammatory response may account for lung damage. This explanation may also apply to the lysophosphatidylcholine, an inflammation promoting compound [16].

The phospholipids appearing at the top of the significantly associated compound list may play a direct causal role as well as an indirect biomarker role in COPD. Their appearance as some of the most significantly associated compounds with FEV_1/_FVC is not surprising given their enrichment overall in the set of significant compounds. Phospholipids, especially sphingomyelin, are consistent with other reports demonstrating association with COPD and related phenotypes, at least in plasma [6]. Causality to COPD may flow from their role in apoptosis, autophagy, cell migration, and cell survival [17]. Alternately, though not mutually exclusive, is the possibility that their presence reflects quantities of plasma membrane derived from dead cells [6].

We clustered all of the data using two strategies, subject level clustering based on similarities across compound profiles (K-means), and compound level clustering based on similarities across subjects (WGCNA). Using BAL, clustering at the subject level differentiated two groups, one with relatively decreased lung function and one with relatively preserved lung function based on FEV_1_/FVC. 

WGCNA was developed for clustering gene-expression profiles, though it has now been adapted to proteomics to find modules associated with a number of diseases including Alzheimer’s, epilepsy, osteoporosis, and lung cancer [18,19,20,21]. In BAL, at the compound level, clustering identified a large module of 1339 compounds, significantly associated with obstructive lung function. Compounds driving the formation of this cluster, those most correlated with the module eigenvector, were individually associated with obstructive lung function. Clustering plasma compounds using these same approaches did not differentiate subjects or compounds by lung function to nearly the same degree. Our results highlight the advantage of detecting COPD associated compounds using BAL as opposed to plasma in this set of subjects. However, previous studies have identified clustered compounds in peripheral blood and observed associations with lung function [3,22]. Potential explanations for the difference in our results with previous work may include the use of a larger sample size (n = 244 vs. our n = 115) [23], use of serum as opposed to plasma [3], incorporating clinical information into clustering [22], using PCA (Principal Component Analysis) as opposed to K-means [22], and comparing advanced disease (GOLD III-IV) versus controls [3].

The large cluster of 1339 co-expressed BAL compounds significantly associated with blood neutrophil count along with the COPD phenotypes. This may reflect the fact that neutrophil count is a possible surrogate for COPD stage [23]. Few compounds in BAL or plasma associated with the cell counts from BAL. Contributing factors may include small sample size (between 70 and 91 of the 115 subjects were matched to BAL cell counts for different types of cell) and measurement technique.

One of the limitations to this study, which occurs in all untargeted metabolomics studies, is that annotation of compounds was limited. Only 343 BAL compounds (4% of total) were annotated with a KEGG ID (201 unique IDs) and 595 plasma compounds (6% of total) were annotated with a KEGG ID (294 unique IDs). As a consequence, pathway analysis was challenging. We attempted to perform enrichment analysis with MetaboAnalyst but were unable to obtain pathway enrichment given our small set of KEGG IDs [24]. As a result, our strategy for identifying enriched categories of compounds amongst those compounds significantly associated with FEV_1_/FVC was to use common, repeated terms found in the chemical names of compounds.

Although this study is very large for a BAL metabolomics study, it may not be large enough to account for the heterogeneity of COPD and to detect compounds with smaller effect sizes. For instance, COPD GWAS (Genome Wide Association Studies) often include tens of thousands of subjects to identify common variants with effect sizes <2. Since BAL is lung fluid and not blood it may be considered closer to the COPD phenotype, justifying a smaller study population than GWAS, though the optimal study size for BAL metabolite studies is not yet clear. Also, the COPD subjects profiled here were mostly mild-to-moderate because very severe subjects were excluded from the bronchoscopy sub-study. 

## 4. Materials and Methods

### 4.1. SPIROMICS

SPIROMICS (ClinicalTrials.gov Identifier: NCT 01969344) is an ongoing multicenter prospective observational study funded by the NIH that enrolled 2982 subjects between November 2011 and January 2015. The institutional review board at all participating sites approved the study protocol (Appendix A). Study participants provided written informed consent (for further details) [12,13]. A subset of 205 subjects participated in a bronchoscopy sub-study as previously described [25]. This study includes 115 subjects that also had simultaneously collected EDTA preserved fresh frozen plasma. All samples underwent quality checks for usability [25,26]. Characteristics of the subjects are shown Table 1. For comparison, characteristics of the SPIROMICS cohort are shown in Appendix A. Our study cohort included 115 subjects, 47 with COPD (FEV_1_/FVC < 0.7 post-bronchodilation), 56 smoking controls, and 12 non-smoker controls (Table 1).

### 4.2. Clinical Variables and Definitions

The following COPD phenotypes were used as outcomes and tested for metabolite associations: % emphysema measured by lung voxels <−950 Hounsfield units at inspiration; postbronchodilator % predicted forced expiratory volume in one second (FEV_1_ %) and ratio of forced expiratory volume in one second to forced vital capacity (FEV_1_/FVC); chronic bronchitis defined as daily productive cough for at least 3 months in the previous two consecutive years; the number of COPD exacerbations leading to hospitalization or requiring antibiotic/corticosteroid treatment in the prior year at baseline visit (exacerbations/yr); clinical covariates (smoking status, current age, sex, and menopause status); whole blood cell counts (neutrophil, lymphocyte, eosinophil, hemoglobin, and hematocrit); and BAL cell counts (macrophages, monocytes, neutrophils, lymphocytes, and eosinophils). Blood and BAL cells were counted using flow cytometry as described in [26]. Not all subjects were matched to BAL cell counts after quality control. BAL cell counts were available for the following number of subjects: eosinophils, 70; neutrophils, 90; lymphocytes, 91; monocytes, 91; and macrophages, 91.

### 4.3. Sample Preparation

Plasma samples were thawed and 100 µL was prepared using methanol precipitation and liquid-liquid extraction as previously described [27]. BAL samples were thawed and prepared with the following modification: 140 µL was aliquoted into a microcentrifuge tube containing 20 µL of internal standards. Samples were vortexed followed by protein precipitation with 560 µL cold methanol, and centrifugation (0 °C for 15 min at 18,000× *g*). The supernatant was removed and placed into two autosampler vials (165 µL for Hydrophilic Interaction Liquid Chromatography (HILIC) and 495 µL for C18 analysis). The samples were dried in a centrifugal evaporator at 45 °C for 2 h. 

The samples for Hydrophilic Interaction Liquid Chromatography HILIC analysis were reconstituted in 30 µL of 95:5 water:acetonitrile. The samples for Reversed phase C18 analysis were reconstituted in 90 µL methanol. 

### 4.4. Liquid Chromatography–Mass Spectrometry—Reversed Phase

Reversed phase samples from the lipid fraction were randomized in the worklist and run randomly in triplicate using an Agilent 1290 series pump with an Agilent Zorbax Rapid Resolution HD (RRHD) SB-C18, 1.8 micron (2.1 × 100 mm) analytical column and an Agilent Zorbax SB-C18, 1.8 micron (2.1 × 5 mm) guard column. The autosampler tray temperature was set at 4 °C, column temperature was set at 60 °C, and the sample injection volume was 8 µL for BAL and 4 µL for plasma. The flow rate was 0.7 mL/min with the following mobile phases: mobile phase A was water with 0.1% formic acid, and mobile phase B was 60:36:4 isopropyl alcohol:acetonitrile:water with 0.1% formic acid. Gradient elution was as follows: 0–0.5 min 30–70% B, 0.5–7.42 min 70–100% B, 7.42–10.4 min 100% B, 10.4–10.5 min 100–30% B, 10.5–15.1 min 30% B. The lipid fraction MS conditions were as follows: Agilent 6545 Quadrupole Time-of-Flight mass spectrometer (QTOF-MS) in positive ionization mode with dual AJS ESI source, mass range 50–1700 m/z, scan rate 2.00, gas temperature 300 °C, gas flow 12.0 L/min, nebulizer 35 psi, sheath gas temperature 275 °C, skimmer 65 V, capillary voltage 3500 V, fragmentor 120 V, reference masses 121.050873 and 922.009798 (Agilent reference mix). 

### 4.5. Liquid Chromatography–Mass Spectrometry—Hydrophilic Interaction

The samples from the aqueous small molecule fraction were analyzed randomly in triplicate using an Agilent 1290 series pump using a Phenomenex Kinetex HILIC, 2.6 µm, 100 Å (2.1 × 50 mm) analytical column and an Agilent Zorbax Eclipse Plus-C8 5 µm (2.1 × 12.5 mm) narrow bore guard column. The autosampler tray temperature was set at 4 °C, column temperature was set at 20 °C, and the sample injection volume was 1 µL for both BAL and plasma. The flow rate of 0.6 mL/min with the following mobile phases: mobile phase A was 50% ACN with pH 5.8 ammonium acetate, and mobile phase B was 90% ACN with pH 5.8 ammonium acetate. Gradient elution was as follows: 0.2 min 100% B, 0.2–2.1 min 100–90% B, 2.1–8.6 min 90–50% B, 8.6–8.7 min 50–0% B, 8.7–14.7 min 0% B, 14.7–14.8 min 0–100% B, 14.8–24.8 min 100% B. The aqueous small molecule fraction MS conditions were as follows: Agilent 6520 QTOF-MS in positive ionization mode with dual ESI source, mass range 50–1700 m/z, scan rate 2.00, gas temperature 325 °C, gas flow 12.0 L/min, nebulizer 30 psi, skimmer 60 V, capillary voltage 4000 V, fragmentor 120 V, reference masses 121.050873 and 922.009798 (Agilent reference mix).

### 4.6. Tandem Mass Spectrometry (MSMS)

The HILIC and C18 LC–MS methods were replicated for tandem MS analysis on the 6520 QTOF and 6545 QTOF, respectively. The MS parameters were adjusted for a scan range 50–1700 m/z, and 10, 20, and 40 eV collision energies with a 500 ms/spectra acquisition time, 1.3 m/z (narrow) isolation width, and 0.25 min delta retention time.

### 4.7. Spectral Peak Extraction

For all datasets, mass spectral peaks were extracted with MassHunter Profinder B.08 SP3 (Build 8137.0) (Agilent) using the “Find by Molecular Feature” algorithm to extract ions above 10,000 counts, followed by the “Find by Ion” algorithm to remine the data by extracting peaks above 8000 counts and filling in missing values. Compounds were aligned across all samples using mass and retention time. The final dataset was exported to Mass Profiler Professional 14.9.1 (MPP, Agilent). In MPP, dilution effects in BAL were corrected based on total useful signal using external scalar. Compounds in all datasets were then identified or putatively annotated.

### 4.8. Compound Identification

Compound identification was performed using IDBrowser in MPP and is based on the current metabolomics standards initiatives (MSI) identification levels. Compound spectra were matched to an in-house developed mass, retention time, and MSMS library build from authentic standards (MSI 1). Compounds not present in the in-house library were identified by matching their MSMS fragmentation spectra to the NIST17 spectral library [28,29] built from reference standards (MSI 2). The remaining unidentified compounds were putatively annotated using accurate mass, chemical formulas, isotope abundance and isotopic distribution to an in-house database comprising METabolite LINk (METLIN), Human Metabolome Database (HMDB), Kyoto Encyclopedia of Genes and Genomes (KEGG) and Lipid Maps. A database score ≥70 out of a possible 100 was considered acceptable for annotation confidence (MSI 3). Compounds that did not match to a name in either the databases or libraries, were subjected to molecular formula generation using the elements C, H, N, O, S, P. All remaining unannotated compounds were designated as a mass@retention time (MSI 4).

### 4.9. Data Processing and Analysis

Unless otherwise mentioned, all analyses were performed with the statistical software package R v3.5.2. Data was preprocessed using the MSPrep R package [30]. A flowchart of data handling is demonstrated in Appendix A. Raw data was split into two parts, one containing compounds with <20% missingness and one containing compounds with missingness greater than this cutoff. K-nearest neighbor imputation (k = 5), using compounds with similar profiles as neighbors, was performed on the compounds with <20% missingness. After imputation compound values were log2 transformed and then batch corrected using Combat [31]. Regression was performed for each compound, regressing clinical outcome and cell counts on a compound feature, compound being the independent variable and clinical outcome or cell count being the dependent variable. Additional covariates in the regression included age, sex, race, asthma, current smoking status, site, chronic bronchitis (Appendix A). Regression was performed only for current and former smokers. Non-smoking controls were used for batch correction, WGCNA module generation, and correlation analysis between BAL and plasma.

Imputation was not performed for compounds with ≥20% missingness, instead the zeros were retained. Non-missing values of compounds with ≥20% missingness were log2 transformed. Then the compound was tested for association with phenotypes using tobit regression [32] (Appendix A).

*P*-values for the coefficient of the compound were controlled for a false discovery rate (FDR) <0.05 using Benjamini–Hochberg correction [33].

### 4.10. Weighted Gene Co-Expression Network Analysis (WGCNA) Technique

WGCNA was used to identify compound modules based on co-expression [34,35]. Thresholds from 1–20 were tested for scale-free topology model fit as demonstrated in the WGCNA tutorial online. Thresholds were chosen to achieve maximum model fit, which resulted in a soft threshold power of six and nine for BAL and plasma respectively. All other parameters were the same for plasma and BAL. The ratio for reassigning compounds between modules was set to zero, the dendrogram cut height for module merging was set at 0.25, and the minimum module size was set at 30. Only the set of compounds with <20% missingness (the set with imputed values) was analyzed with WGCNA per the package authors’ recommendations [36]. Pearson correlation between the first eigenvector from each WGCNA module and the clinical variables was used to determine significance of association between clinical variables and modules.

### 4.11. Clustering

K-means clustering using squared Euclidian distances was used to identify groups of subjects with compound profiles that were similar. Silhouette scores were assessed for k = 2 through 10 clusters. The greatest average silhouette width for the different cluster numbers was used to decide the optimal number of k clusters.

### 4.12. Classification of Compounds

Name annotation of compounds was performed in MPP (14.9.1). We used repeated terms found in the chemical names of compounds to classify molecules by type. Each name, when one was available, was searched for common organic compound terms. Regular expressions representing compound types are shown in Appendix A. Names containing the search term were categorized as containing the search term compound. Enrichment of molecule types within sets of significant (based on association with clinical variable tested) was based on the Fisher’s exact test.

## 5. Conclusions

This study demonstrates that BAL and plasma reflect distinct aspects of the COPD metabolome; the plasma metabolome is more strongly associated with age, sex, and red cell counts, while BAL has strong associations with spirometry, current smoking, emphysema, and neutrophil counts. Some of the classes of BAL compounds that were associated with COPD phenotypes included: phosphatidylethanolamines, phosphatidylcholines, amino acid derived compounds (most significantly arginine, lysine, leucine, isoleucine, and serine), and fatty acids. Similar to the transcriptome we also found that cell counts are strongly associated with the metabolome, suggesting that clinical metabolomics studies should include cell counts in their regression models.

The metabolome differences are also reflected by the small number of WGCNA modules in BAL as compared with plasma, along with the largest BAL module correlating significantly with FEV_1_/FVC. FEV_1_/FVC ratios, while clustering of subjects based on their plasma compound profile did not. 

## Figures and Tables

**Figure 1 metabolites-09-00157-f001:**
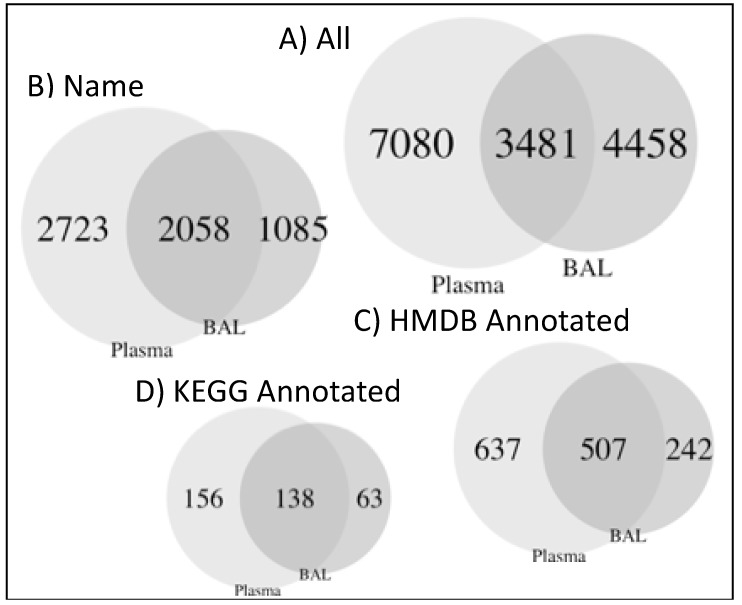
Compilation of Venn diagrams for bronchoalveolar lavage (BAL) and plasma compounds. All compounds (**A**), only annotated compounds (**B**), or annotated compounds with identifiers in HMDB (**C**), or KEGG (**D**) databases.

**Figure 2 metabolites-09-00157-f002:**
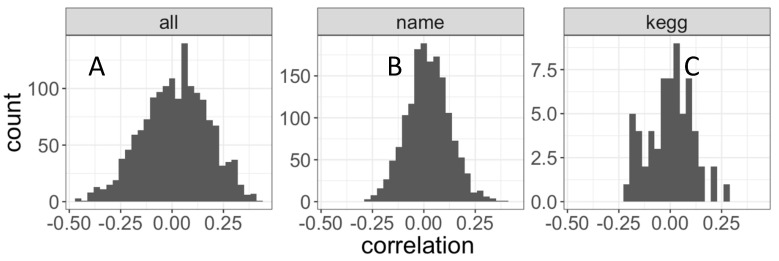
BAL and Plasma comparison using distribution of Pearson’s correlation between BAL and plasma. All compounds (left), only annotated compounds (middle), or annotated compounds with KEGG identifiers (right). Mean correlation and t-test *p*-values are mean = 0.015, *P* = 1.7 × 10^−4^ (**A**), mean = 0.021, *P* = 4.7 × 10^−15^ (**B**) and mean = 1.1 × 10×, *P* = 0.94 (**C**). The set of all compounds passing the <20% missingness preprocessing filter and annotated with HMDB identifiers was equivalent to the corresponding KEGG set, and no separate distribution for HMDB is shown.

**Figure 3 metabolites-09-00157-f003:**
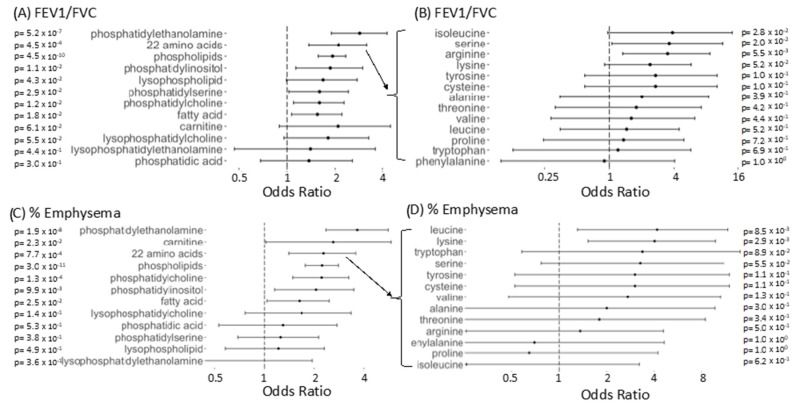
Enrichment of compound classes in BAL compounds associated with FEV_1_/FVC and % Emphysema). (**A**) Odds ratio with 95% confidence intervals for compounds in a given category to appear among the FDR corrected FEV_1_/FVC associated compounds versus appearing among non-associated compounds, using Fisher’s exact test. Regular expression searches identified compounds of different categories with matching compounds checked manually for accurate categorization. (**B**) Same as A (top left) but for more specific amino acid containing compounds. Categories shown in B include amino acid containing compounds for amino acids with >10 compounds detected experiment-wide for BAL. In B, a compound need only contain the compound listed to be included. (**C**) Same as A for % Emphysema (**D**) Same as B for % Emphysema

**Figure 4 metabolites-09-00157-f004:**
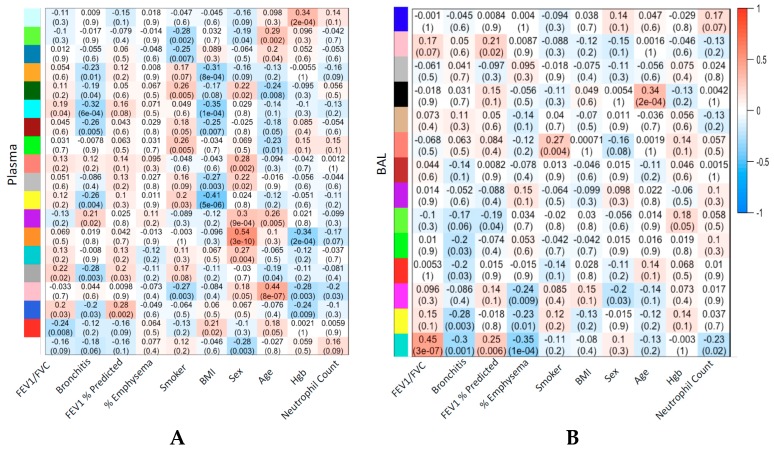
Strength of correlation between weighted gene co-expression network analysis (WGCNA) modules and clinical variables and outcomes. Heatmap of module/clinical variable correlation. Plasma (**A**) or BAL (**B**). Module colors correspond to dendrogram in Appendix A. Cell text is Pearson correlation (*p*-value) between the first principal component representing the module and the corresponding variable. Plasma WGCNA modules without any correlation, *p*-values <0.01, are excluded (12 excluded, 19 displayed) for greater visual clarity. Full plasma WGCNA module to clinical variable correlations are shown in Appendix A.

**Table 1 metabolites-09-00157-t001:** Cohort characteristics.

Variable	Non-Smokers	Smoking Controls	COPD	*p*-Value
n	12	56	47	
Sex, % men	33	45	62	0.104
Race,% White	50	73	87	7.48 × 10^−3^ *
Race, % Black	25	21	6	7.48 × 10^−3^ *
Race, % Asian	17	2	4	7.48 × 10^−3^ *
Race, % other	8	4	2	7.48 × 10^−3^ *
Age, yr	56 (50–60)	58 (50–66)	64 (58–68)	7.95 × 10^−4^ *
Current smokers, %	0	36	36	2.68 × 10^−2^ *
Pack–years	0 (0–0)	34 (26–44)	42 (34–60)	3.95 × 10^−11^ *
Body mass index	26.21 (5.46)	28.78 (4.47)	28.9 (5.27)	0.198
Chronic bronchitis, %	0 (0)	7 (26)	15 (36)	0.294
Exacerbations/yr	0.08 (0.29)	0.12 (0.43)	0.39(0.68)	0.117
Emphysema, %	0.15 (0.06–1.22)	0.16 (0.05–0.4)	1.05 (0.32–2.5)	2.90 × 10^−3^ *
FEV_1_ %	99.29 (7.31)	100.23 (13.1)	78.97 (19.92)	3.87 × 10^−8^ *
FEV_1_/FVC	81 (77–87)	78 (75–81)	61 (55–67)	5.31 × 10^−24^ *

Data are presented as median (interquartile range) or mean ± SD. * *p*-value < 0.05. Emphysema, %: % Emphysema voxels (<−950 Hounsfield units) in lung CT (Computed Tomography) image. Exacerbations/yr: # of exacerbations in last year. Chronic bronchitis: Daily productive cough for at least 3 months in the previous 2 consecutive years. % FEV_1_: Postbronchodilator % predicted forced expiratory volume in one second. FEV_1_/FVC: Ratio of forced expiratory volume in one second to forced vital capacity. COPD: chronic obstructive pulmonary disease.

**Table 2 metabolites-09-00157-t002:** Significantly associated compounds with clinical, cell-type, and COPD sub-phenotype variables.

Variable	BAL	Plasma
Sex	1	240
Current Smoker	249	7
Age	0	177
Menopause	0	0
Neutrophil Count	665	0
Lymphocyte Count	5	0
Eosinophil Count	0	4
BAL Neutrophil Count	0	4
BAL Lymphocyte Count	1	0
BAL Eosinophil Count	0	7
BAL Monocyte Count	1	0
BAL Macrophage Count	1	1
Hemoglobin	0	63
Hematocrit	0	80
FEV_1_/FVC	1230	0
Emphysema, %	791	2
Chronic Bronchitis	0	0
Exacerbations/yr	1	0
FEV_1_ %	8	0

Cells are populated with numbers obtained after testing all compounds, 7939 from BAL and 10,561 from plasma. Compound measures with >20% missingness in the raw data were modeled using tobit regression. Compound measures ≤20% missingness in the raw data were modeled using either beta, logistic, negative binomial, or linear regression (Appendix A). Compounds were significant at *p*-value adjusted FDR (False Discovery Rate) <0.05 Emphysema, %: % Emphysema voxels (<−950 Hounsfield units) in lung CT image; FEV_1_ %: Postbronchodilator % predicted forced expiratory volume in one second; FEV_1_/FVC: Ratio of forced expiratory volume in one second to forced vital capacity; Exacerbations/yr: # of exacerbations in last year.

**Table 3 metabolites-09-00157-t003:** Compounds in BAL most significantly associated with FEV_1_/FVC with corresponding plasma results.

Compound	FDR BAL	Estimate BAL	SE BAL	FDR Plasma	Estimate Plasma	SE Plasma
PS (37:3)	7.6 × 10^−5^	0.45	0.089	1	0.0015	0.094
Lophocerine	7.6 × 10^−5^	0.42	0.084	1	−0.0034	0.066
p-cresol	7.6 × 10^−5^	0.4	0.08	0.98	−0.036	0.14
PE (38:3)	7.6 × 10^−5^	0.38	0.075	0.93	0.086	0.094
PC (40:6)	7.6 × 10^−5^	0.35	0.069	0.11	0.14	0.033
PC (40:6) (isomer)	7.6 × 10^−5^	0.34	0.063	0.68	−0.16	0.079
Ceramide (d18:1/16:0) *	7.6 × 10^−5^	−0.29	0.054	0.89	0.092	0.086
PC (32:1) **	7.6 × 10^−5^	0.28	0.054	0.96	−0.048	0.082
Glycocholic acid *	7.6 × 10^−5^	0.27	0.052	0.96	0.023	0.035
MGDG (36:5)	7.6 × 10^−5^	0.27	0.055	0.89	21	20
S-(Phenylacetothiohydroximoyl)-L-cysteine	7.6 × 10^−5^	0.26	0.051	0.78	−0.13	0.09
SM (d18:1/24:1) **	7.6 × 10^−5^	0.26	0.051			
PE (35:1)	7.6 × 10^−5^	0.26	0.05	0.96	−0.036	0.075
N-palmitoyl glycine	7.6 × 10^−5^	0.25	0.05	0.92	17	20
L-Threonylcarbamoyladenylate	7.6 × 10^−5^	0.25	0.049	0.55	−0.078	0.033
Decaprenyl phosphate	7.6 × 10^−5^	0.24	0.047	0.99	−2.9	11
Mycalamide B	7.6 × 10^−5^	0.23	0.044	0.97	−0.0099	0.027
PC (36:4) *	7.6 × 10^−5^	0.23	0.046	0.44	36	14
PE (36:3)	7.6 × 10^−5^	0.22	0.045	0.96	0.019	0.042
PC (34:2) **	7.6 × 10^−5^	0.22	0.044	0.95	5.9	8.5
Homocysteine *	7.6 × 10^−5^	0.22	0.046	0.89	1.6	1.4
SQMG (16:1)	7.6 × 10^−5^	0.21	0.042	0.55	−26	12
PE (34:2) *	7.6 × 10^−5^	0.2	0.039	0.98	−0.019	0.081
CL (70:0)	9.2 × 10^−5^	0.27	0.056	0.98	−0.015	0.071
CL (72:7)	9.4 × 10^−5^	0.40	0.082	1	0.001	0.11

Top 25 compounds for BAL FEV_1_/FVC association after sorting of the FDR *p*-value and estimate. * indicates an accurate mass and retention time match, ** indicates an accurate mass and MSMS library match. SE: Standard Error; FDR: False discovery rate based on Benjamini–Hochberg; CL: cardiolipin; SM: Sphingomyelin; PC: Phosphatidylcholine; PE: Phosphatidylethanolamine; PS: Phosphatidylserine; SQMG: Sulfoquinovosyl monoacylglycerol; MGDG: Monogalactosyldiacylglycerol.

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
