# Peer review of "Bronchoalveolar Lavage Fluid from COPD Patients Reveals More Compounds Associated with Disease than Matched Plasma"

_metabolites, 2019, doi:10.3390/metabo9080157_

Round 1
Reviewer 1 Report
The manuscript of Halper-Stromberg et al. (entitled Bronchoalveolar lavage fluid from COPD patients reveals more compounds associated with disease than plasma) compares plasma and BALF samples in a COPD-subgroup of a previously published larger cohort by a network-based metabolomics approach in order to find potential biomarkers, associated with disease phenotypes.
The subject of this work would be of interest and the present manuscript fits to the previous works of the authors in this field.
The experimental procedures and the statistics seem to be appropriate.
The results are clearly presented, the major conclusions are convincing.
The conclusion is solid but clearly shows the added value of the paper, although the major aspects are associations and/or correlations, without further explanation.
Altogether, the reviewer congratulates to the authors for their work, and suggests some minor points to improve the paper.
Major point(s): none
Minor points:
1. Minor or major blood contamination makes the analysis of BALF samples difficult. The authors should provide data, containing at least blood biomarker (e.g. free haemoglobin in BALF), that the BALF samples, investigated in their study, are free of haemorrhagic bias.
2. Surprisingly the authors found lysoPC(17:0), as one of the compounds, most significantly associated with FEV1/FVC (Table 3). LysoPC(17:0) is a quite unusual minor lysophospholipid, containing of a margaric acid chain at the C-1 position, derived from milk, butter or fat of ruminating animals. In this context lysoPC(17:0) may reflect the nutritional habit of study-probands. How do the authors explain the occurence of lysoPC(17:0) in BALF (and not the more common lysoPC species with 16, 18 or 20 carbon atoms), and its high correlation with FEV1/FVC?
3. The paper contains several abbreviations, which are not explained correctly. For example, CL(70:0) and CL(72:7) in Table 3. Are these cardiolipins?
Please, check the whole manuscript for such abbreviations.
4. Literature-22 is incompletely cited, volume and page numbers are missing.
5. Literature-37 is incompletely cites, at least the URL of this Web-based tutorial must be given.
Reviewer 2 Report
This is the first report regarding the biochemicals present in the Bronchoalveolar Lavage Fluid From COPD Patients, and I, not only me, supposedly other many readers, would be surprised by such findings.
The findings given by the authors are rather leading in this field, the important things are the authors data may be further developed as the preclinical signals (or markers) which may become the relevant leading markers in the preventive medicines.
